# Truss Metamaterials: Multi-Physics Modeling for Band GapTuning

Daniel Calegaro *⬤ and Stefano Mariani ⬤

Dipartimento di Ingegneria Civile e Ambientale, Politecnico di Milano, Piazza Leonardo da Vinci, 32, 20133 Milano, Italy; stefano.mariani@polimi.it
* Correspondence: daniel.calegaro@polimi.it

**Abstract:** Periodic elastic metamaterials (EMMs) display the capability to forbid the transmission of elastic waves for certain frequency ranges, leading to band gaps. If topology optimization strategies are exploited to tune the band gaps of EMMs, the said band gaps cannot be modified in real-time. This limitation can be overcome by allowing for active materials in the design of EMMs. In this work, a hyperelastic piezoelectric composite was considered to assess the coupled effects of material and geometric nonlinearities on the behavior of sculptured microstructures featuring a three-dimensional periodicity. Specifically, it was assumed that the composite material is obtained by embedding piezo nanoparticles into a soft polymeric matrix. In this way, piezoelectricity and instability-induced pattern transformation could be fully exploited to actively tune the band gaps. A thermodynamically consistent multi-physics model for the active composite material is discussed and implemented in a general-purpose finite-element code. The reported results of the simulations showed how the band gaps are affected by the aforementioned nonlinearities and by a feature of the architected periodic cell linked to its topology.

**Keywords:** elastic metamaterials; hyperelasticity; piezoelectricity; buckling

## 1. Introduction

Waves represent a pervasive form of motion in nature. The study of wave propagation and tuning based on material properties may substantially contribute to the advancement of science and technology. In particular, tailoring the propagation of sound and elastic waves is of main concern in fields such as medical [1], military [1], automotive [2], and civil engineering [3,4]. More specifically, the propagation of elastic waves can be forbidden within specific frequency ranges, called band gaps, thanks to the absorption of the elastic wave energy.

Many natural materials possess the ability to control waves and prevent their propagation. However, these materials alone cannot be used for the purpose of reaching the desired acoustoelastic properties, due to poor control of their physical properties; new artificial materials look, therefore, necessary to solve this problem. Taking advantage of composites with architected unit cells at the microscale, a new class of man-made materials has been devised, the so-called elastic metamaterials (EMMs). As far as the materials are concerned, a number of studies investigated the effects induced by the contrast in Young's modulus, Poisson's ratio, and the density between the matrix and inclusions in the case of a binary composite [5,6]. On the other hand, as far as the topology is concerned, which indeed represents one of the most-important features affecting the band gap properties, research has been conducted at the unit cell level, focusing on the lattice parameters to attain an optimal design [7–9] through topology and parametric optimization techniques [10–13]. For instance, in [14,15], an optimization method was proposed on the basis of a closed-form estimation of the band gap width and of the starting frequency as a function of a number of key geometric parameters; such an approach resulted in being useful in obtaining the optimal bang gap and the material design, to achieve better properties.

In order to extend even more the tunability of the EMM properties, active materials have been proposed so that the band gaps can be modulated by the application of external non-mechanical stimuli. Recent works include the exploitation of material nonlinearities and geometrical nonlinear effects, such as buckling instabilities, to tune the band gaps by switching the pattern of deformation [16–19]. Furthermore, the adoption of multi-field couplings was studied to understand the effect of different modulation techniques related to electric and magnetic biasing [20,21], temperature [22], and piezoelectricity [23–25].

Within the realm of piezoelectric materials, a promising solution involves the use of three-dimensional periodic architected truss cells, in place of the more-common two-dimensional periodic structures. This method may entail the manufacturing of architected periodic cells with piezoelectric nanoparticles embedded in a polymeric matrix [26]. By tuning the topology of the unit cell, a range of piezoelectric properties can be achieved, even larger than those obtained by bulk piezoelectric materials, supplemented by increased flexibility.

Here, we propose the use of an EMM composite, truss-like, periodic unit cell to appropriately tune its band gaps. Owing to the soft (compliant) nature of the composite due to its polymeric matrix, we move from the work of Guo [27], where a generalized framework for an electro-mechanical coupling in the presence of finite deformations was developed, to model the response of the composite cell. By including a hyperelastic energy contribution into the electric enthalpy density function, a thermodynamically consistent multi-physics model was implemented in the finite-element software COMSOL Multiphysics® [28]. The proposed tuning of the band gap properties was obtained by taking advantage of the interplay of: (1) geometric nonlinear effects, in terms of buckling instabilities; (2) material nonlinearities, arising from the hyperelastic piezoelectric model; (3) the tuning of the lattice parameters of the unit cell, or representative volume. Such a strategy is revealed to be effective in largely increasing the number of available degrees of freedom to control the propagation of the elastic waves in the EMM.

The remainder of this paper is arranged as follows. In Section 2, the thermodynamically consistent multi-physics formulation to model the soft piezoelectric material is provided; as this model has been ad hoc implemented in COMSOL Multiphysics®, the details are given in order to understand how it can cope with all the nonlinearities allowed for in the study. Results regarding a specific architected unit cell are discussed in Section 3, to see how band gaps can be tuned also by means of the triggered microscopic instabilities. Some concluding remarks and foreseen future work directions are finally discussed in Section 4.

## 2. Materials and Methods

To properly introduce the coupled electro-mechanical model adopted for piezoelectricity in finite deformations, a Lagrangian description is proposed departing from the conservation of energy. In rate form, the electro-mechanical power per unit volume in the reference configuration [29,30] reads:

$$\dot{U} = S_{ij}\dot{\varepsilon}_{ij} + E_i\dot{D}_i, \tag{1}$$

where an indicial notation has been adopted; therefore, indices $i, j, k, l, m = 1, 2, 3$ are used to represent the components of the tensors coming into play in a three-dimensional orthonormal reference frame, and a superposed dot stands for the time rate. In Equation (1): $U$ is the stored energy density; $S_{ij}$ the second Piola–Kirchhoff stress tensor; $\varepsilon_{ij}$ the Green–Lagrange strain tensor; $E_i$ the electric field; and $D_i$ the electric displacement field. By defining the electric enthalpy density $H$ as:

$$H = U - E_iD_i, \tag{2}$$

its time rate is obtained from Equation (1) as:

$$\dot{H} = S_{ij}\dot{\varepsilon}_{ij} - D_i\dot{E}_i. \tag{3}$$

Hence, $H$ is a function of the Green–Lagrange strain tensor and of the electric field, namely $H = H(\varepsilon_{ij}, E_i)$. The stress and electric displacement relations, thus, read:

$$S_{ij} = \frac{\partial H}{\partial \varepsilon_{ij}}, \tag{4}$$

$$D_i = -\frac{\partial H}{\partial E_i}, \tag{5}$$

where, as implicitly assumed in Equation (1), the measure of stress and strain considered in the formulation is conjugate in energy. The piezoelectric constitutive relations can be then obtained by defining a specific form of $H$. It can be assumed to be [27,31]:

$$H = \Psi^{ME}(C_{ij}) - \mathcal{E}_{ikl}\varepsilon_{kl}E_i - \frac{1}{2}\xi J C_{ij}^{-1} E_j E_i, \tag{6}$$

where: $\Psi^{ME}(C_{ij})$ is the stored mechanical energy density; $C_{ij} = 2\varepsilon_{ij} + \delta_{ij}$ is the right Cauchy–Green tensor; $\delta_{ij}$ is the Kronecker delta; $\mathcal{E}_{ikl}$ is the piezoelectric coupling tensor; $\xi$ is the dielectric permittivity constant; and $J$ is the determinant of the deformation gradient. In this specific case, the second Piola–Kirchhoff stress tensor and the electric displacement field become, respectively:

$$S_{ij} = \frac{\partial \Psi^{ME}}{\partial \varepsilon_{ij}} - \mathcal{E}_{kij}E_k - \frac{1}{2}\xi J \left( C_{kl}^{-1}C_{ij}^{-1} - 2C_{ki}^{-1}C_{lj}^{-1} \right) E_k E_l, \tag{7}$$

$$D_i = \mathcal{E}_{ikl}\varepsilon_{kl} + \xi J C_{ij}^{-1} E_j. \tag{8}$$

Due to its intrinsic characteristics, the piezoelectric effect is usually investigated under infinitesimal deformations. In fact, crystals and some ceramics are the most-widespread piezoelectric materials, so elastic constitutive equations in the small deformation range prove sufficient to model the mechanical backbone of the piezoelectric effect. By borrowing the relevant linear relation, in the finite deformation range, a Kirchhoff-like strain energy density can be assumed for $\Psi^{ME}$, which reads:

$$\Psi^{ME} = \frac{1}{2}K_{ijkl}\varepsilon_{ij}\varepsilon_{kl}, \tag{9}$$

where $K_{ijkl}^E$ is the elasticity tensor. Hence, the sole mechanical stress $S_{ij}^{ME}$ takes the form:

$$S_{ij}^{ME} = \frac{\partial \Psi^{ME}}{\partial \varepsilon_{ij}} = K_{ijkl}\varepsilon_{kl}. \tag{10}$$

This relation, together with Equations (7) and (8), provides a generalization to the considered deformation regime of the standard e-form of piezoelectric constitutive equations [30]. In view of the piezoelectric nanoparticles embedded in a soft matrix, in the present investigation, we instead employed a form of the internal mechanical energy density more appropriate for polymeric materials, i.e., a nonlinear hyperelastic one. Specifically, the Neo-Hookean strain energy density is assumed for simplicity:

$$\Psi^{ME} = \frac{1}{2}\mu \left[ tr(C_{ij}) - 3 \right] - \mu \ln J + \frac{1}{2}\Lambda(\ln J)^2, \tag{11}$$

where $\Lambda$ and $\mu$ are the relevant Lamé constants. Hence, the mechanical stress relation becomes:

$$S_{ij}^{ME} = \frac{\partial \Psi^{ME}}{\partial \varepsilon_{ij}} = 2\frac{\partial \Psi^{ME}}{\partial C_{ij}} = \mu \delta_{ij} + (\Lambda \ln J - \mu)C_{ij}^{-1}. \tag{12}$$

This phenomenological piezoelectric constitutive model has been implemented in COMSOL Multiphysics® through the MEMS module, to investigate the band gap properties of the architected unit cell shown in Figure 1. Specifically, the model implementation has been made possible by enabling the *Equation View* COMSOL Multiphysics® option, in order to access and modify the equations and variables used internally by the software.

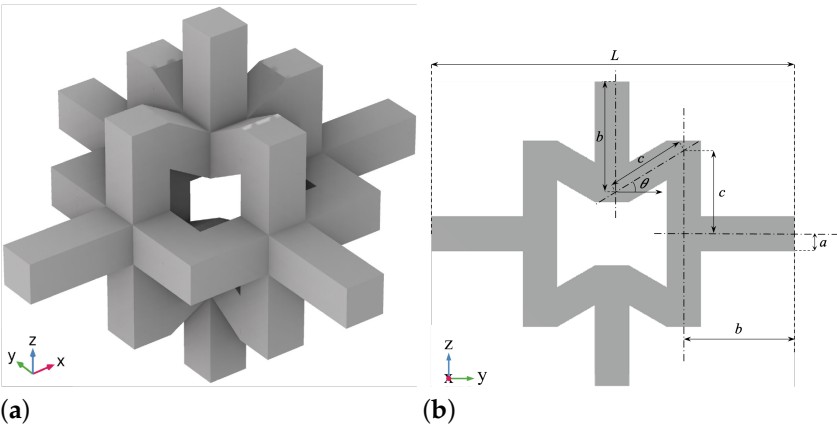

**Figure 1.** (**a**) Three-dimensional model of the unit cell structure, and (**b**) cross-section with the principal dimensions characterizing its geometry.

By varying the applied uniaxial compression in the *z* direction, the said interaction among the material and geometric nonlinearities in the model may provide a means to tune the band gaps and, therefore, exploit the architected composite for different real-world applications. The unit cell, depending on the angle $\theta$ denoting the slope of the internal out-of-plane beams, may result in displaying an auxetic response. The entire numerical investigation includes four steps: (1) a Bloch–Floquet instability analysis, (2) a linear buckling analysis, (3) a post-buckling analysis, and (4) a final wave propagation analysis in the buckled configuration.

The Bloch–Floquet instability analysis is carried out to determine at which level of the macroscopic average compressive strain an instability can be encountered. Depending on the strain level, two main types of mechanical instabilities can emerge [17]: the microscopic one, with a wavelength comparable to the dimension of the unit cell; the macroscopic one, with a wavelength much larger than the size of the unit cell. In the analyses, two steps are necessary to define the wavelength, and so the size, related to the instability. First, general periodic boundary conditions are imposed on the unit cell in order to statically deform it, and periodic boundary conditions in terms of the electric potential are assumed at the boundary. Second, Bloch periodic boundary conditions in terms of displacement and electric field are imposed on the unit cell to solve the eigenfrequency problem for a specific set of wave vectors. Such Bloch boundary conditions take the form:

$$u_i(\mathbf{x} + \mathbf{r}, t) = u_i(\mathbf{x}, t)e^{\gamma \mathbf{k} \cdot \mathbf{r}}, \tag{13}$$

where: $u_i$ is the displacement field; $\mathbf{x}$ is the spatial coordinate vector; $\mathbf{r}$ is the distance vector in the current configuration between pairs of nodes placed on the opposite sides of the unit cell; $t$ is the time; $\mathbf{k}$ is the Bloch wave vector; $\gamma$ is the imaginary unit; and $u_i(\mathbf{x}, t) = \tilde{u}_i(\mathbf{x})e^{-\gamma \omega t}$, $\omega$ being the relevant angular frequency. The compressive strain, concretely enforced in the *z* direction by means of a macroscopic or effective strain in the same direction is gradually increased until the eigenfrequency becomes zero, meaning that a transition between a stable configuration (characterized by real eigenfrequency values) to an unstable configuration (characterized by complex eigenfrequency values) has occurred [17,32].

Once the size of the microscopic instability has been identified through the aforementioned Bloch–Floquet instability analysis, a linear buckling analysis is performed on an

enlarged unit cell to obtain the geometry of the first buckling mode. Such an enlarged unit cell is set according to Bertoldi et al. [17]. On the basis of the deformed pattern, a post-buckling analysis is carried out by imposing a further effective deformation up to the desired strain level. By setting the path of the wave vector, a series of wave propagation analyses was finally conducted at different strain levels, to see how the band gaps may change by varying the deformation.

The dimensions used to characterize the primitive periodic unit cell in Figure 1 are $a = 1.55$ cm, $b = 6.43a$, and $c = 4.84a$. The electromechanical properties of the composite material were taken from the work of Cui et al. [26], so they read: Lamé constants of the Neo-Hookean model $\Lambda = 65{,}792$ MPa and $\mu = 257$ MPa; non-vanishing components of the stress–charge coupling tensor $\mathcal{E}_{113} = \mathcal{E}_{123} = 0.0336$ C/m$^2$, $\mathcal{E}_{311} = 0.00310$ C/m$^2$, $\mathcal{E}_{322} = 0.00850$ C/m$^2$, and $\mathcal{E}_{333} = 0.0291$ C/m$^2$; dielectric permittivity constant $\xi = 0.2576 \times 10^{-9}$ C$^2$/(Nm$^2$); mass density $\rho = 1360$ kg/m$^3$. As concerns the mesh adopted for the analyses, hexahedral elements were used with a characteristic size small enough to address the nonlinear response of the architected unit cell, as observed with a convergence analysis at varying mesh density.

## 3. Results and Discussion

The results of the Bloch–Floquet instability analysis are reported in Figure 2. Specifically, Figure 2a shows how the values of the critical stretch $\lambda^{cr}$ at the onset of the microscopic and macroscopic instabilities are affected by the angle $\theta$ in the range $0°$–$45°$. It can be seen that, moving from the undeformed configuration characterized by $\lambda = 1$, microscopic instability (red line) always occurs before the macroscopic one (blue line). This means that, for this specific architected cell, the band gap properties can be adjusted through a compression-induced microstructural modification. Conversely, if the macroscopic instability took place first, the microstructure would not change and, so, even the properties of the band gaps. It is also interesting to note that the plane in which the microscopic instability is preferentially activated is the *y-z* one; see Figure 2f. This outcome is basically ruled by the difference in the magnitudes of the entries $\mathcal{E}_{311}$, $\mathcal{E}_{322}$, and $\mathcal{E}_{333}$ of the stress–charge coupling tensor. In the results, the two components $\mathcal{E}_{322}$ and $\mathcal{E}_{333}$ with a larger amplitude enforce the said plane in which the instability has to occur.

Figure 2b shows instead an exemplary nominal stress vs. the applied stretch response, as obtained with the post-buckling analysis of the periodic cell featuring $\theta = 30°$. Looking at the pattern evolution in Figure 2c–f, the trusses or beams in the unit cell can be classified as follows, on the basis of their deformation. External beams of length $b$, which are referred to as elastic ligaments, connect adjacent primitive unit cells and display a lower rigidity against the type of deformation pattern induced by the triggered microinstability. Internal beams of length $c$, which are considered as masses, display instead a higher stiffness against the instability-induced deformation pattern. The pattern evolution testifies to the fact that the initial cell response is characterized by uniform compression up to the critical stretch that triggers the microscopic instability. Next, as a consequence of the aforementioned microscopic instability, the deformation gets localized in the elastic ligaments, while the masses show a tendency to rigid body-like rotations only. A remarkable softening is, thus, observed, as evidenced by the change in the slope of the stress–strain curve in Figure 2b.

We now focus on the overall effects of microscopic instability and nonlinear polymer behavior on the band gap properties. Two different tuning methods were investigated, by varying the applied stretch along the *z* direction and the angle $\theta$; the results obtained through the eigenfrequency analysis are reported in Figure 3 in terms of the band gap plots for $\theta = 35°$, considering both the undeformed state ($\lambda = 1$) and the deformed one ($\lambda = 0.9$). In these diagrams, the normalized frequency $\tilde{f} = \omega L/(2\pi c_{t0})$, where $c_{t0} = \sqrt{\mu/\rho}$ is the speed of shear waves in the bulky polymer, is reported as a function of the wave vector **k**, which is varied along the path $\Gamma - X - M - \Gamma - Z - R - A - Z$, depicted in Figure 3a. As can be seen, the width and position of the band gaps are constant along the entire path connecting the high symmetry points, at varying values of the stretch $\lambda$. Therefore, in

the following analysis, where the stretch and the angle $\theta$ were, respectively, varied in the range 0.9–1 and 0°–45°, the wave vector **k** was considered to change only along the path $\Gamma - X - M - \Gamma$. The corresponding results of the analyses are shown in Figure 4a–j.

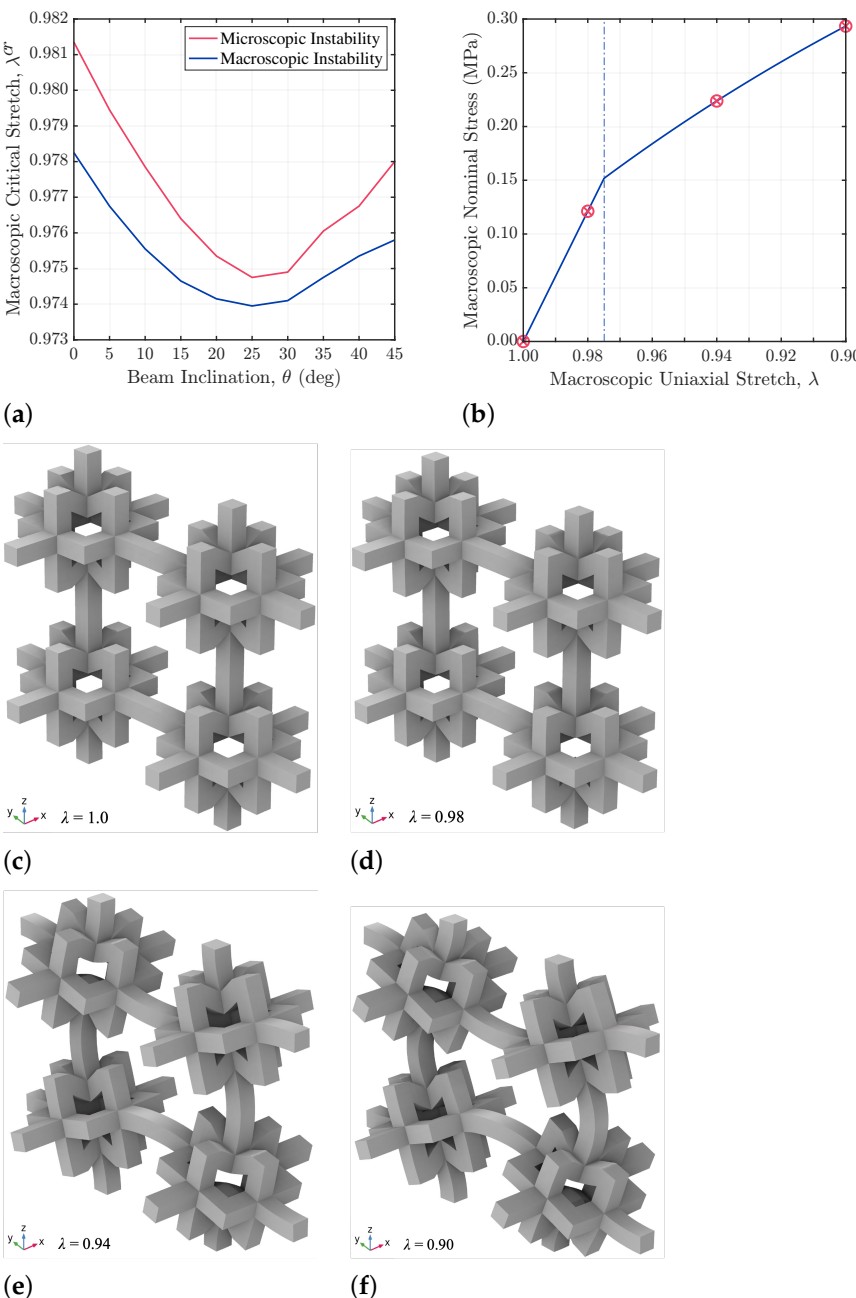

**Figure 2.** Pattern evolution and microscopic/macroscopic instability at the unit cell level: (**a**) critical stretch $\lambda^{cr}$ for microscopic and macroscopic instabilities at varying beam slope $\theta$; (**b**) macroscopic nominal stress vs. applied stretch response of the enlarged unit cell ($\theta = 30°$), where the vertical dotted line corresponds to the inception of microscopic instability and the red circles denote the states reported in (**c**–**f**), to gain insights into the microscopic instability-induced pattern evolution.

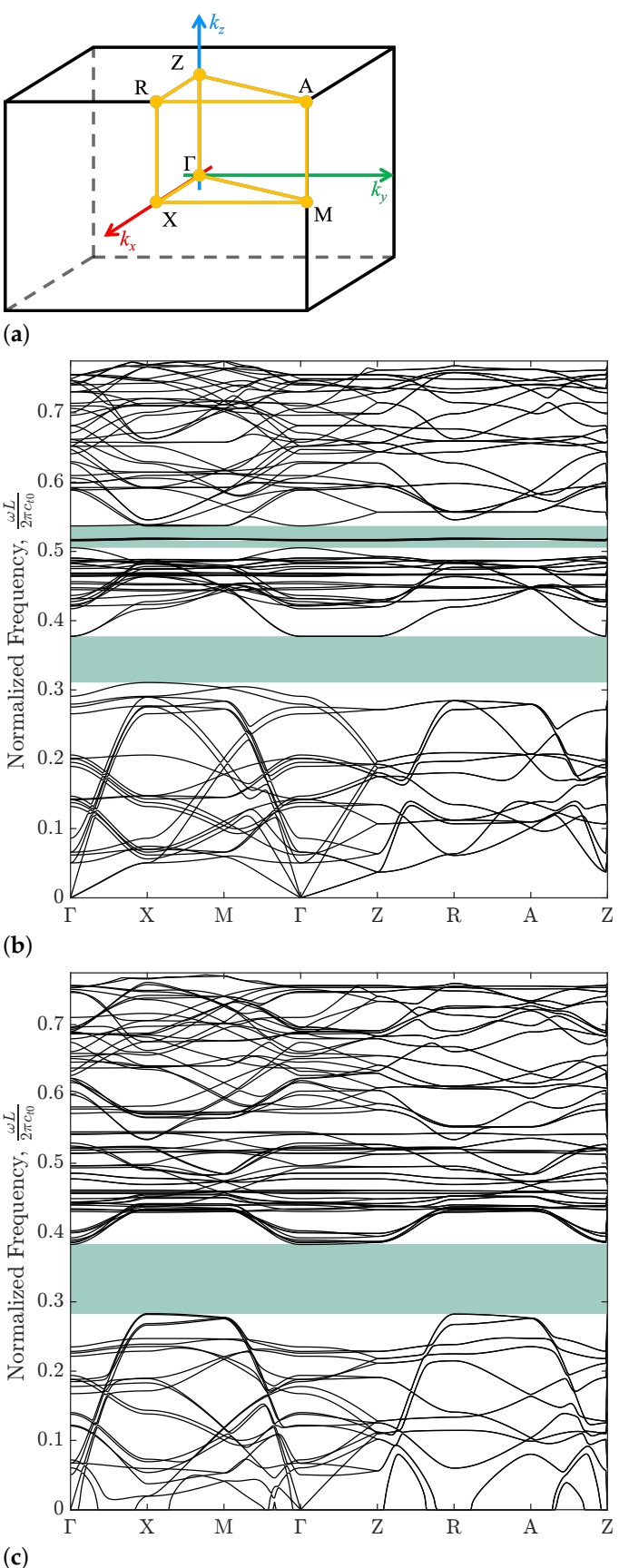

**Figure 3.** (**a**) First irreducible Brillouin region. Band gap plots of the enlarged unit cell with a beam slope $\theta = 35°$ and loaded in the $z$ direction, for (**b**) $\lambda = 1$ and (**c**) $\lambda = 0.9$.

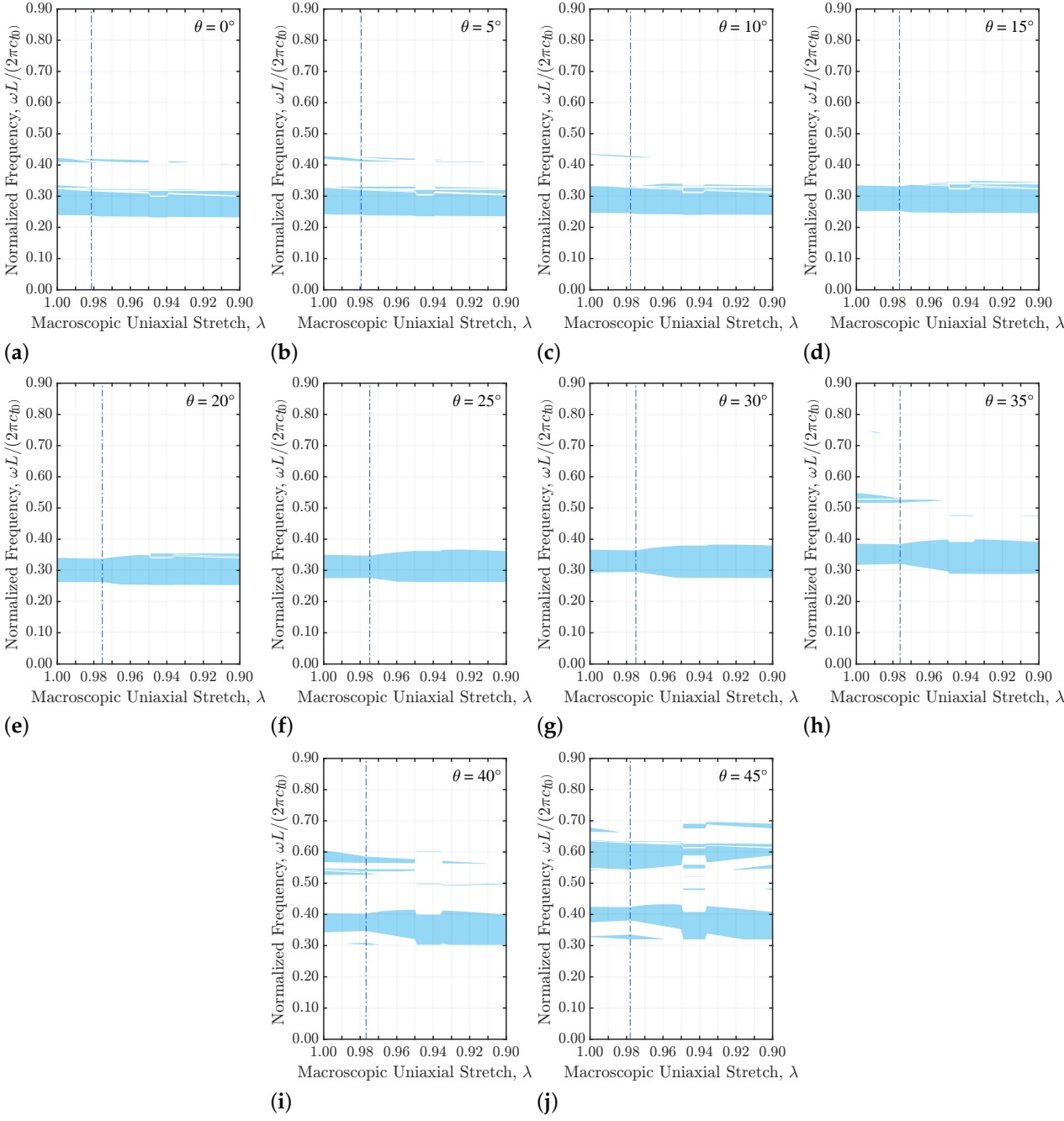

**Figure 4.** (**a**–**j**) Joint effects of the stretch $\lambda$ and beam slope $\theta$ on the band gaps (shaded light-blue regions) of the enlarged unit cell loaded in the $z$ direction. As in Figure 2, the vertical dotted lines denote the critical stretch at the inception of microscopic instability.

At fixed beam slope $\theta$, the sole effect produced by the pattern-induced instability on the band gap properties can be observed. Such an effect becomes more relevant after the critical stretch for the microscopic instability has been attained, and the major band gap typically increases in size. If the angle $\theta$ is instead increased, at fixed applied stretch $\lambda$, all the reported band gaps shift toward higher frequencies since the overall stiffness of the structure increases as well. When both $\theta$ and $\lambda$ grow, the average frequency of the lower band gap tends to decrease, as clearly visible in the figure for the cases featuring $\theta = 35°$, $\theta = 40°$, and $\theta = 45°$. Moreover, for these high $\theta$ values, the low-frequency band gaps are less prone to be split into smaller band gaps; the other way around, additional band gaps appear at higher frequencies.

## 4. Conclusions

In this paper, we investigated the exploitation of material and geometric nonlinearities, together with the multi-physics modeling of smart materials, to tune the band gap properties of architected periodic elastic metamaterials.

The present model was developed in a fully thermodynamically consistent way, moving from the electric enthalpy density of a piezoelectric material to account for finite deformation effects (nonlinear geometrics). The mechanical contribution to the said enthalpy was selected in a form appropriate for compressible Neo-Hookean materials, due to the soft hyperelastic behavior of the considered polymer matrix. The tunability of the frequency band gap properties was analyzed, allowing for instability-induced pattern transformations that occur at increasing compressive strains applied to the primitive unit cell of the truss metamaterial. A specific geometric feature of the architected cell topology, namely the slope $\theta$ of the internal beams, was varied to to assess the effects on the mentioned tunability.

The results showed an expansion of the band gaps with the increase of the applied compressive strain. Furthermore, for high $\theta$ values, the mid-frequency of the lower band gaps decreased. On the other hand, by increasing $\theta$ at fixed applied strain, the band gaps shifted toward higher frequencies. These findings demonstrate the potential benefits of jointly exploiting geometric and material nonlinearities to dynamically adjust the frequency band gaps.

In conclusion, the results presented in this work open up new avenues for further research and development in the field of active metamaterials. The findings will be further developed to consider poling-induced anisotropy in the response of the architected unit cell and verified through laboratory tests on 3D-printed prototypes of the proposed active metamaterials. It is foreseen that more-effective and -versatile active metamaterials, with improved tunability and functionality, can be the backbone of innovative applications in various fields.

**Author Contributions:** Conceptualization, D.C. and S.M.; methodology, D.C. and S.M.; formal analysis, D.C. and S.M.; investigation, D.C. and S.M.; resources, S.M.; data curation, D.C. and S.M.; writing—original draft preparation, D.C. and S.M.; writing—review and editing, S.M.; supervision, S.M. All authors have read and agreed to the published version of the manuscript.

**Funding:** This research received no external funding.

**Data Availability Statement:** The data presented in this study are available upon request from the corresponding author.

**Conflicts of Interest:** The authors declare no conflict of interest.

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
