# Peer review of "Truss Metamaterials: Multi-Physics Modeling for Band GapTuning"

_machines, doi:10.3390/machines11090913_

Round 1

Reviewer 1 Report

The communication manuscript entitled “Truss metamaterials: multi-physics modelling for band gaps tuning”, by Calegaro D. and Mariani S., used hyperelastic piezoelectric composite to assess the coupled effects of material and geometric nonlinearities on the behaviour of sculptured microstructures featuring a three-dimensional periodicity. It is a well-established work, and its novelty could attract the interest of the Machines' readers. However, I would like to ask the authors to provide more clarification in the discussion section and expand it more.

it is fluent and understandable.

Author Response

We would like to thank the reviewers for their careful reviews. We modified the text to reply to their comments. In addressing the comments and suggestions, main changes are highlighted in red in the new version of the manuscript.

Here below you will find our replies to all the comments.

Reviewer #1

The communication manuscript entitled “Truss metamaterials: multi-physics modelling for band gaps tuning”, by Calegaro D. and Mariani S., used hyperelastic piezoelectric composite to assess the coupled effects of material and geometric nonlinearities on the behaviour of sculptured microstructures featuring a three-dimensional periodicity. It is a well-established work, and its novelty could attract the interest of the Machines' readers.

We thank the reviewer for the positive comment.

However, I would like to ask the authors to provide more clarification in the discussion section and expand it more.

As requested by the other reviewers too, we expanded Section 3 to add additional results. The band gap plots relevant to the undeformed and deformed configurations for the exemplary geometry characterized by q=35° are now reported in the new Figure 3. As already partially commented in the text, results do not change significantly with the geometry and we adopted these specific plots to highlight the insensitivity of the band gaps on the wave vector (with either in-plane or out-of-plane orientation). These graphs provide now the support for the comments to follow, related to the band gaps only.

Reviewer 2 Report

In the proposed paper, the Authors illustrate the multi-physics modelling of centimetric elastic metamaterials. The analysis shows both microscopic and macroscopic instabilities that raise from the y-z plane deformation of the structure, and they affect the bandgap. The manuscript could be considered for the publication, but some considerations should be carried out, even as a response to the following questions:

- could the Author provide more details about the methods for the finite element simulations (exploited modules, boundary conditions,etc)? 

- What is the effect of the deformation on the wave propagation along the other dimensions? For a complete description, The Authors should illustrate the response in frequency in the x and y directions. 

- The Authors highlight the tunability of the device. Could the torque applied (expecially to reach the extreme case of theta=45°) result in the irreversible modification of the whole structure? In this frame and for the considered band, the oscillation frequencies could compromise the active tunability of the array? 

- could the Authors provide a comparison with devices with similar purpose and dimensions? Otherwise the reader could not find the novelty of the present work.

The Authors stated 'The findings will be further developed to consider poling-induced anisotropy in the response of the architected unit cell and verified through laboratory tests on 3D-printed prototypes of the proposed active metamaterials.'.

Despite the Authors intentions, it is useless to anticipate the planning of further studies. 

Author Response

We would like to thank the reviewers for their careful reviews. We modified the text to reply to their comments. In addressing the comments and suggestions, main changes are highlighted in red in the new version of the manuscript.

Here below you will find our replies to all the comments.

Reviewer #2

In the proposed paper, the Authors illustrate the multi-physics modelling of centimetric elastic metamaterials. The analysis shows both microscopic and macroscopic instabilities that raise from the y-z plane deformation of the structure, and they affect the bandgap. The manuscript could be considered for the publication, but some considerations should be carried out, even as a response to the following questions:

- could the Author provide more details about the methods for the finite element simulations (exploited modules, boundary conditions,etc)? 

In Section 2, we added some comments to explain how Comsol Multiphysics was used in order to obtain the results gathered in this work. Regarding the boundary conditions adopted at each stage of the analysis, not only for the mechanical problem, they are discussed extensively in Section 2 starting at line 119.

- What is the effect of the deformation on the wave propagation along the other dimensions? For a complete description, The Authors should illustrate the response in frequency in the x and y directions. 

In order to explain the results in detail, we added Figure 3 and the relevant comments in the text of Section 3.

- The Authors highlight the tunability of the device. Could the torque applied (expecially to reach the extreme case of theta=45°) result in the irreversible modification of the whole structure? In this frame and for the considered band, the oscillation frequencies could compromise the active tunability of the array? 

In the present work, the material is assumed to always behave (hyper)elastically. That means that no dissipation takes place at the material level, due to e.g. irreversible phenomena like plasticity or fracture. The interaction between instability and plasticity, or even cracks that can modify the topology of the unit cell, is a very complex matter which is out of the scope of the present investigation. In the authors’ opinion, it could be understood if coupled with experimental data to fully characterized the response of soft piezoelectric materials.

- could the Authors provide a comparison with devices with similar purpose and dimensions? Otherwise the reader could not find the novelty of the present work.

In the Introduction, we briefly discussed the findings by other works in this field, to account for the pattern of deformation and the multiphysics governing the problem, see references [7-23]. The specific results could not be compared one-to-one to those available in the literature, since the geometry has been specifically devised to display the characteristic response described in Section 3. The main reference could be paper [24], which was indeed of inspiration to develop the thermodynamically consistent material model, but was reporting results with a different aim.

The Authors stated 'The findings will be further developed to consider poling-induced anisotropy in the response of the architected unit cell and verified through laboratory tests on 3D-printed prototypes of the proposed active metamaterials.'.

Despite the Authors intentions, it is useless to anticipate the planning of further studies. 

Reviewers typically ask to add foreseen activities as a follow up of the investigation, and we therefore included this statement which is indeed something we are currently trying to pursue. If the reviewer is convinced that it does not add much to the discussion and could be also misleading, we can drop it.

Reviewer 3 Report

machines-2565781

Truss metamaterials: multi-physics modelling for band gaps tuning

The manuscript develops a theoretical analysis of truss metamaterials using software COMSOL. Basically, the investigation is interesting and worthing to be published. Several problems in the manuscript need to be revised:

1. More details about the computational procedures by using software COMSOL should be given.

2. There are currently many acoustical metamaterials, such as phononic crystals (DOI 10.1142/S0217984922500105; 10.3390/ma13092106; 10.1109/ACCESS.2019.2946085; 10.3390/cryst7110328, etc.), and the advantages and innovations of the proposed method are not well described.

3. Why giving the first several formulas? Are they connecting to the usage of software COMSOL?

Author Response

We would like to thank the reviewers for their careful reviews. We modified the text to reply to their comments. In addressing the comments and suggestions, main changes are highlighted in red in the new version of the manuscript.

Here below you will find our replies to all the comments.

Reviewer #3

The manuscript develops a theoretical analysis of truss metamaterials using software COMSOL. Basically, the investigation is interesting and worthing to be published. Several problems in the manuscript need to be revised:

  1. More details about the computational procedures by using software COMSOL should be given.

As requested also by the other reviewers, we added some comments to explain how Comsol Multiphysics was used in order to obtain the results gathered in this work

  1. There are currently many acoustical metamaterials, such as phononic crystals (DOI 10.1142/S0217984922500105; 10.3390/ma13092106; 10.1109/ACCESS.2019.2946085; 10.3390/cryst7110328, etc.), and the advantages and innovations of the proposed method are not well described.

We thank the reviewer for bringing to our attention these papers. We added them in the Introduction and we also discussed the main ideas in them.

  1. Why giving the first several formulas? Are they connecting to the usage of software COMSOL?

As stated, we implemented the thermodynamically consistent material model in Comsol Multiphysics and, since there are no specific references where this formulation can be found in the implemented fashion, we decided to take the description in it. This goes in the direction of providing to the readers all the details to be able to reproduce the same results, which would be impossible without a thorough description of the constitutive law for the material under different stimuli.

Round 2

Reviewer 2 Report

The responses of the Authors satisfied the raised suggestions and comments, and quality of the proposed paper was improved. Therefore, the article could be considered for publication.